# The value of red blood cell distribution width, neutrophil-to-lymphocyte ratio, and hemoglobin-to-red blood cell distribution width ratio in the progression of non-small cell lung cancer

Jin-liang Chen[1☯], Jin-nan Wu[2☯], Xue-dong Lv[1], Qi-chang Yang[3], Jian-rong Chen[1]*, Dong-mei Zhang[4]*

**1** Department of Respiratory Medicine, The Second Affiliated Hospital of Nantong University, Nantong, Jiangsu, People's Republic of China, **2** Postgraduate in Respiratory Medicine, The Second Affiliated Hospital of Nantong University, Nantong, Jiangsu, People's Republic of China, **3** Department of Pathology, The Second Affiliated Hospital of Nantong University, Nantong, Jiangsu, People's Republic of China, **4** Department of Medical Research Center, The Second Affiliated Hospital of Nantong University, Jiangsu, People's Republic of China

☯ These authors contributed equally to this work.
* drchenjr@163.com (JC); zdm_ntyy@163.com (DZ)

**Data Availability Statement:** All relevant data are within the manuscript and its Supporting Information files.

## Abstract

### Background

Lung cancer is the leading cause of cancer-related deaths worldwide, with non-small cell lung cancer (NSCLC) accounting for 85% of all lung cancer cases. Inflammation has been proven to be one of the characteristics of malignant tumors. Chronic inflammatory response mediated by cytokines in the tumor microenvironment is an important factor in tumorigenesis. The purpose of this study was to observe and evaluate the value of red blood cell distribution width (RDW), neutrophil-to-lymphocyte ratio (NLR), and hemoglobin-to-red blood cell distribution width ratio (HRR) in the progression of NSCLC.

### Methods

A total of 245 patients with NSCLC, 97 patients with benign pulmonary nodules, and 94 healthy volunteers were included in this study. Factors, such as age, gender, smoking history, histological type, lymph node metastasis, distant metastasis, TNM stage, and differentiation degree were statistically analyzed. The correlation of RDW, NLR, and HRR of patients with NSCLC with other clinical experimental parameters were also analyzed. Then, the diagnostic value of RDW, NLR, and HRR in the progression of NSCLC was evaluated.

### Results

RDW, NLR, and HRR could be used to distinguish patients with NSCLC from healthy controls (p < 0.05). In addition, only the RDW in the NSCLC group with III-IV stage was significantly different from that in the benign pulmonary nodules group (p = 0.033), while NLR and

**Funding:** Jin-liang Chen: the Natural Science Foundation of Jiangsu Province (BK20191207); Jin-nan Wu: Postgraduate Research & Practice Innovation Program of Jiangsu Province (SJCX19_0873); Dong-mei Zhang: the Scientific Research Project of Health Commission of Jiangsu Province (H2018035); the Scientific Research Project of "333 Project" in Jiangsu Province (BRA2018224); Jian-rong Chen: the Science and Technology Program of Nantong City (Grant No. HS2018002). Jin-liang Chen: Conceptualization, Methodology, Software, Investigation; Jin-nan Wu: Formal analysis, Data curation, Software, Investigation, Writing-Original draft preparation; Xue-dong Lv: Visualization, Investigation; Qi-chang Yang: Supervision; Dong-mei Zhang: Software, Validation; Jian-rong Chen: Project administration, Writing-Reviewing and Editing. The authors thank all the staff members in our institution.

**Competing interests:** The authors have declared that no competing interests exist.

**Abbreviations:** NSCLC, non-small cell lung cancer; RDW, red blood cell distribution width; NLR, neutrophil-to-lymphocyte ratio; HRR, hemoglobin-to-red blood cell distribution width ratio; CEA, carcinoembryonic antigen; ROC, receiver operating characteristic; AUC, area under curve; CI, confidence interval; VEGF, vascular endothelial growth factor; HIF, hypoxia-inducible factor.

HRR could significantly distinguish patients with NSCLC and benign pulmonary nodules (p < 0.001). RDW and NLR were positively correlated with NSCLC stage, whereas HRR was negatively correlated with NSCLC stage. RDW, NLR, and HRR were also significantly associated with the differentiation degree of NSCLC (p < 0.05). The ROC curve analysis showed that the combination of RDW with NLR, HRR, and CEA could show significantly higher diagnostic value than any one marker alone (AUC = 0.925, 95% CI: 0.897–0.954, and sensitivity and specificity of 79.60% and 93.60%, respectively).

## Conclusion

RDW, NLR, and HRR can be utilized as simple and effective biomarkers for the diagnosis and evaluation of NSCLC progression.

## Introduction

Lung cancer is the leading cause of cancer-related deaths worldwide and has the highest death rate among all malignancies [1]. Non-small cell lung cancer (NSCLC) accounts for about 85% of all lung cancer cases [2]. The vast majority of patients with NSCLC were diagnosed in the middle and advanced stages, and their 5-year overall survival rate was less than 20% [3]. This is mainly because the late diagnosis and poor treatment of NSCLC. Inflammation has been proven to be the eighth feature of malignant tumors [4]. Chronic inflammatory response mediated by cytokines in the tumor microenvironment is an important factor in tumorigenesis [5]. Considering that human lungs are susceptible to a variety of poisons and pathogens, chronic injuries and inflammation may occur, which may cause lung cancer [6].

Routine blood test for lung cancer is a prerequisite for the hospital admission of patients. Neutrophils are a type of white blood cells that play an important role in the defense and protection of the body. Lymphocytes are another type of white blood cells and an quite important component of the body's immune response. Neutrophils and lymphocytes have been shown to be widely involved in cancer progression [7–9]. Studies have demonstrated that neutrophil-to-lymphocyte ratio (NLR) is a simple and easily accessible inflammation biomarker that can be potentially used to predict cancer progression and prognosis [10–12]. Red blood cell distribution width (RDW) mainly reflects the heterogeneity of red blood cell volume, and is usually applied to diagnose and distinguish types of anemia. Recent studies have found that RDW is related to the systemic inflammatory response and nutritional status of the body, and this relationship may have implications for tumor occurrence and development in the body [12–14]. Hemoglobin-to-red blood cell distribution width ratio (HRR) is a new biomarker. HRR was first proposed to have predictive value for cancer by Sun et al. [15]. They found that HRR had important implications in evaluating the prognosis of patients with esophageal squamous cell carcinoma. In addition, HRR has been reported in other types of tumors in recent years, such as head and neck tumors and small cell lung cancer [16,17].

Therefore, the purpose of this study was to conduct a retrospective study based on routine clinical blood test results of patients with NSCLC to explore and evaluate whether RDW, NLR, and HRR can be used as simple, cheap and effective biomarkers for the diagnosis and assessment of clinical progression conditions in NSCLC.

## Materials and methods

### 1. Patients

A total of 245 patients with NSCLC who were admitted to the Second Affiliated Hospital of Nantong University from June 2016 to June 2019 were diagnosed through lung tissue or lymph node biopsy, pleural effusion cytology and pathological examination. The participants had not received radiochemotherapy, immune-targeted therapy or surgery prior to selection. A total of 97 patients with benign pulmonary nodules who were admitted to the Second Affiliated Hospital of Nantong University at the same time were selected and confirmed by histopathology. These patients had no history of acute and chronic infectious diseases, no history of malignant tumors, and no important systemic organ diseases. In addition, 94 volunteers who underwent physical examination in the same period were selected as the healthy control group. The healthy volunteers had no acute and chronic infectious diseases; no vital organ diseases; and no genetic family tumor history. The details of the number of people in each group are shown in Fig 1. Tumor staging was performed in accordance with the lung cancer TNM staging guidelines published by the Union for International Cancer Control in 2018. This study was approved by the Medical Ethics Committee of the Second Affiliated Hospital of Nantong University. All patients signed the informed written consent. The electronic medical record system was used to collect baseline characteristics, including patient age, gender, smoking status, performance status, and medical history, as well as the statistics and analysis of clinical blood routines, tumor screening, and other test data.

### 2. NLR and HRR calculations

The following equations were used: NLR = neutrophil count ($\times 10^9$/L) / lymphocyte count ($\times 10^9$/L), HRR = hemoglobin (g / L) / red blood cell distribution width (%).

### 3. Statistical analysis

All data were analyzed using the statistical software IBM SPSS Statistics 22.0 and the graphing software Graphpad Prism 8.0. All data were tested by Kolmogorov-Smirnov test or Shapiro-Wilk test to determine whether they were normally distributed. For normally distributed data, the t-test was used to compare the mean between two groups with equal variances. When the variance was not the same, the t'test (Satterthwaite method) was used. For non-normally distributed data, use the non-parametric test (Mann-Whitney U test), and described as the median value (interquartile range) [M (Q1-Q3)]. Count data were compared using the $\chi 2$ test. Correlation analysis between two variables was performed using linear correlation analysis. Pearson correlation analysis was used for data that corresponded to bivariate normal distribution, and Spearman correlation analysis was used for data that fitted non-parametric distribution. The diagnostic value of RDW, NLR and HRR for NSCLC was analyzed using ROC curves. $p < 0.05$, the difference was statistically significant.

## Results

### 1. Baseline characteristics of all participants

A total of 245 patients with NSCLC were included in this study. Among these patients, 107 had squamous cell carcinoma and 138 had adenocarcinoma. In addition, 71.43% of the patients with NSCLC were in TNM III-IV stage, and 68.16% of the patients had poorly differentiated tumors (Table 1). The study also included 97 patients with benign pulmonary nodules and 94 healthy volunteers. No difference in age and sex was observed among the three groups,

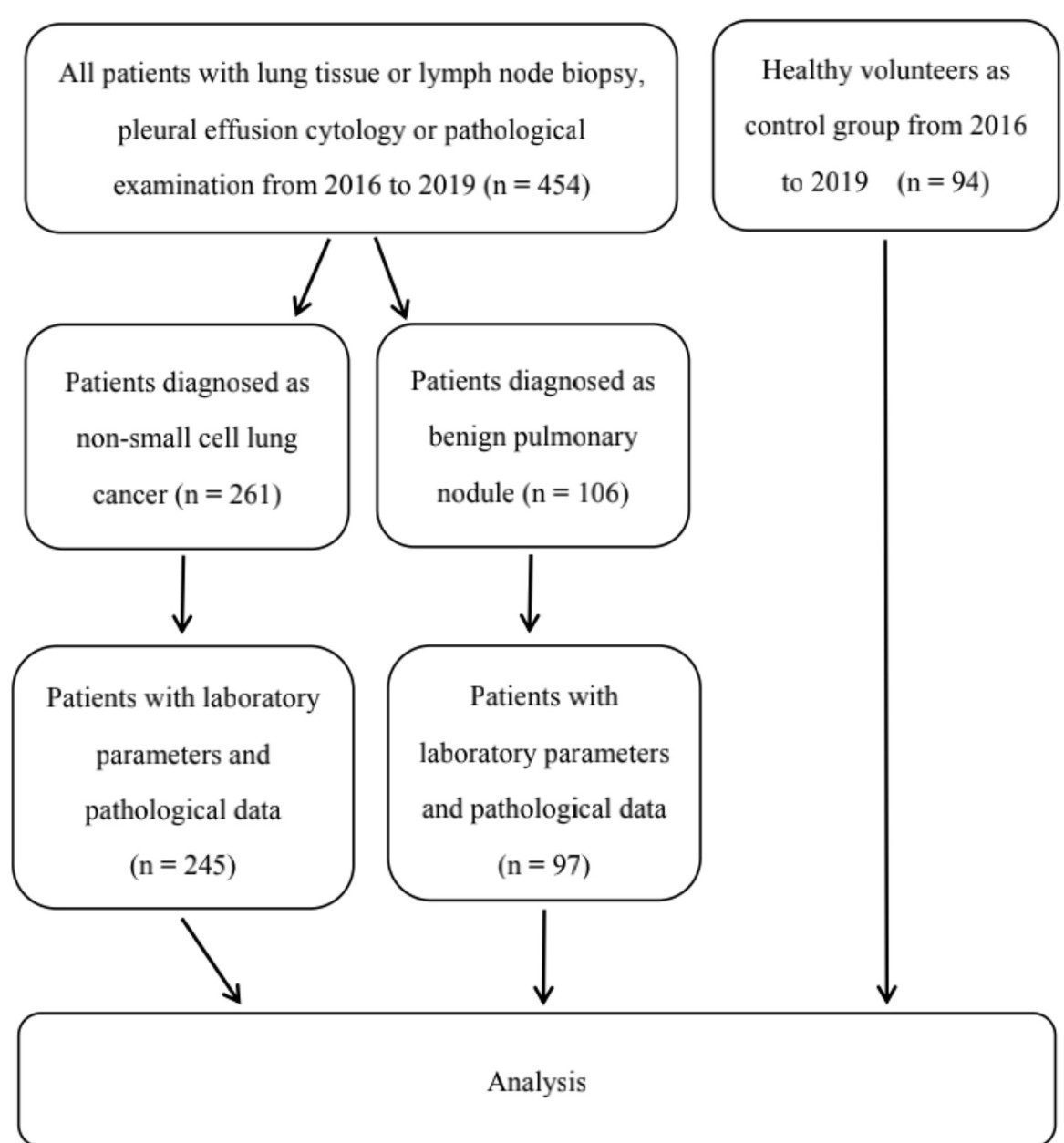

**Fig 1. The flow chart of the study.**

but between NSCLC patients and benign pulmonary nodule patients, smoking history ($p < 0.05$) was closely related (Table 2).

## 2. Expression levels of RDW, NLR, and HRR in the NSCLC patient group, benign pulmonary nodules patient group, and healthy volunteer group

Leukocyte count, neutrophil count, and CEA in patients with NSCLC were significantly higher than those in patients with benign lung nodules and healthy controls (Table 2). The median of RDW in the group of patients with NSCLC was 13.4%, which was significantly higher than

Table 1. Demographic and clinical characteristics of NSCLC patients(n = 245).

| Variables | NSCLC patients |
|---|---|
| Age (years) | 66(58–73) |
| Male n(%) | 132(53.88%) |
| Smoking history n(%) | 134(54.69%) |
| Squamous cell carcinoma n(%) | 107(43.67%) |
| Lymph node metastasis n(%) | 156(63.67%) |
| Distant metastasis n(%) | 129(52.65%) |
| TNM III-IV stage n(%) | 175(71.43%) |
| Poorly differentiation n(%) | 167(68.16%) |

Data were expressed as number (%) and median (interquartile range, 25th–75th).

*NSCLC* non-small cell lung cancer.

that in the healthy control group (12.9%) (p < 0.05) (Table 2). However, compared with the RDW of the patients with benign pulmonary nodules, that of NSCLC patients with I-II stage was no statistically different, whereas that of the patients with III-IV stage was significantly different (p = 0.033) (Fig 2A). We then further calculated the NLR and HRR values of each group. As can be seen in Table 2, the NLR of the group of patients with NSCLC was significantly higher than that of the healthy control group and the benign pulmonary nodule group (p < 0.001). By contrast, the level of HRR in patients with NSCLC was significantly lower than that in patients with benign pulmonary nodules and healthy volunteers (p < 0.001).

In addition, we also found that NSCLC patients with I-II stage had significantly lower RDW values than patients with III-IV stage (p = 0.022, Fig 2A). Similarly, the NLR value of NSCLC patients with III-IV stage was significantly higher than that of patients with I-II stage (p < 0.001, Fig 2B). By contrast, for the levels of HRR, NSCLC patients with III-IV stage were significantly lower than patients with I-II stage (p < 0.001, Fig 2C). In addition, compared with the healthy control group, patients with benign pulmonary nodules did not have significantly different levels of NLR and HRR (Fig 2B and 2C), but had significantly different RDW (p = 0.038, Fig 2A).

The correlation of NSCLC stage with RDW, NLR, and HRR is shown in Fig 3A–3C. Correlation analysis showed that NLR was positively correlated with NSCLC stage (r = 0.212, p < 0.001), whereas HRR was negatively correlated with the cancer stage (r = -0.233, p < 0.001). In addition, there was a very weak correlation between RDW and NSCLC stage (r = 0.128, p = 0.045).

## 3. Correlation between clinicopathological features and RDW, NLR, and HRR in 245 NSCLC cases

ROC curves were drawn on the basis of the RDW, NLR, and HRR values of patients with NSCLC and healthy volunteers. The Youden 's index (sensitivity + specificity -1) was used to determine the cut-off value. Results showed that the optimal cut-off values of RDW, NLR, and HRR were 13.25, 2.14, and 9.48, respectively, and the area under the curve (AUC) of RDW, NLR, and HRR were 0.629, 0.846, and 0.696, respectively (Fig 4). Therefore, we divided the patients with NSCLC into two groups in accordance with the optimal cut-off values of RDW, NLR, and HRR, and then analyzed the relationship of these parameters with the clinicopathological characteristics of the patients. The results were shown in Table 3. The RDW ≥ 13.25 group had more patients with NSCLC with age ≥64 years (p = 0.002), squamous cell carcinoma (p = 0.037), TNM III-IV stage (p = 0.022), and poor tumor differentiation (p = 0.004) than the RDW < 13.25 group. Similarly, compared with those in the NLR < 2.14 group, patients with

**Table 2. Baseline characteristics of all participants.**

| Variables | Normal range | NSCLC patients | Benign lung nodules patients | Healthy volunteers | P value |
|---|---|---|---|---|---|
| Number | | 245 | 97 | 94 | |
| Clinical characteristics | | | | | |
| Age (years) | - | 66(58–73) | 65(56–69) | 64(49.75–78.00) | 0.052 |
| Gender (male/female) | - | 132/113 | 53/44 | 40/54 | 0.062 |
| Smoking history(yes/no) | - | 134/111[a] | 38/59 | 27/67[c] | <0.05 |
| Laboratory parameters | | | | | |
| Leukocytes, ×$10^9$/L | 3.5–9.5 | 6.40(5.35–8.10)[a] | 5.20(4.35–6.00)[b] | 5.70(4.90–6.43)[c] | <0.05 |
| Neutrophils, ×$10^9$/L | 1.8–6.3 | 4.30(3.31–5.85)[a] | 2.80(2.40–3.50) | 3.15(2.70–3.85)[c] | <0.001 |
| Lymphocytes, ×$10^9$/L | 1.1–3.2 | 1.40(1.00–1.80)[a] | 1.80(1.40–2.10)[b] | 1.95(1.60–2.23)[c] | <0.05 |
| Hemoglobin, g/L | 115–150 | 132(117–141)[a] | 140(132–152) | 137.5(131–151.3)[c] | <0.001 |
| RDW, % | 10.0–15.0 | 13.4(12.90–14) | 13.2(12.80–13.85)[b] | 12.9(12.70–13.50)[c] | <0.05 |
| NLR | - | 3.21(2.21–5.34)[a] | 1.73(1.51–2.08) | 1.63(1.33–2.00)[c] | <0.001 |
| HRR | - | 9.853(8.67–10.85)[a] | 10.48(9.85–11.45) | 10.75(9.78–11.57)[c] | <0.001 |
| CEA, ng/mL | - | 4.06(2.53–16.63)[a] | 2.02(1.49–2.55)[b] | 1(0.6–1.93)[c] | <0.001 |

Data were expressed as number and median (interquartile range, 25th–75th).

Binary logistic regression analysis with adjustment age and gender was used to control confounding factors.

*RDW* red blood cell distribution width; *NLR* neutrophil-to-lymphocyte ratio; *HRR* hemoglobin-to-red blood cell distribution width ratio; *CEA* carcinoembryonic antigen; *NSCLC* non-small cell lung cancer.

P values were calculated by Kruskai-Wallis tests.

[a] Indicates a significant difference (P < 0.05) between NSCLC patients and Benign lung nodules patients.

[b] Indicates a significant difference (P < 0.05) between Benign lung nodules patients and Healthy volunteers.

[c] Indicates a significant difference (P < 0.05) between NSCLC patients and Healthy volunteers.

NSCLC in the NLR ≥ 2.14 group had higher age (p = 0.012), larger tumor tissue (p = 0.002), higher degree of lymph node metastasis (p = 0.002), farther distant metastases (p < 0.001),

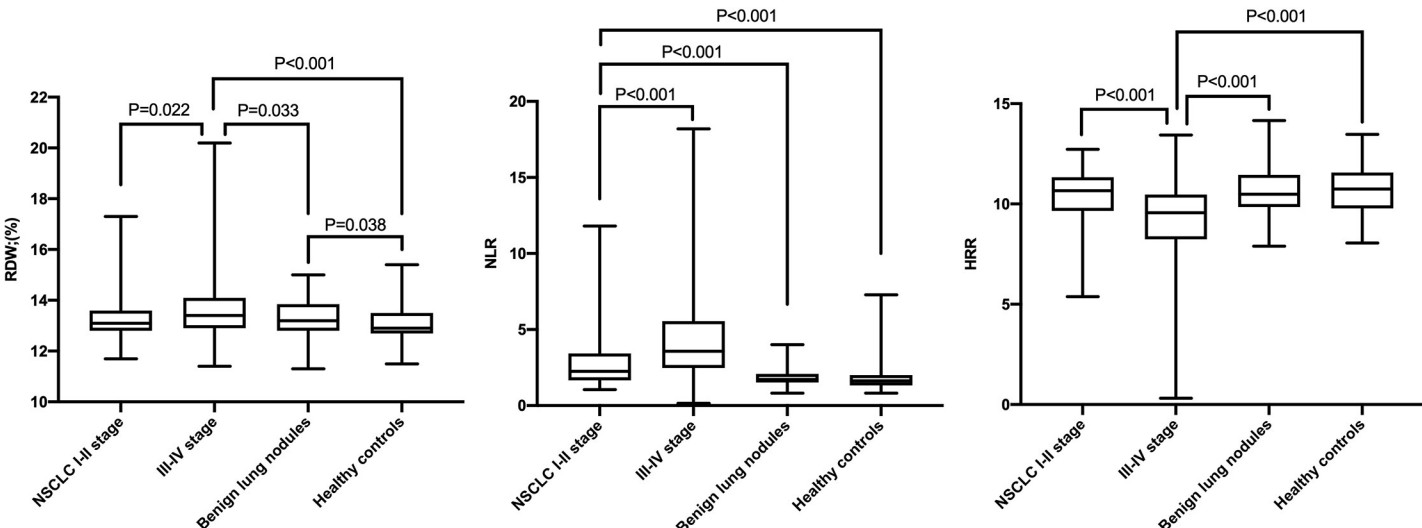

**Fig 2. A.** Expression level of RDW in NSCLC group, benign pulmonary nodules group, and healthy controls group. RDW red blood cell distribution width; NSCLC non-small cell lung cancer. **B.** Expression level of NLR in NSCLC group, benign pulmonary nodules group, and healthy controls group. NLR neutrophil-to-lymphocyte ratio; NSCLC non-small cell lung cancer. **C.** Expression level of HRR in NSCLC group, benign pulmonary nodules group, and healthy controls group. HRR hemoglobin-to-red blood cell distribution width ratio; NSCLC non-small cell lung cancer.

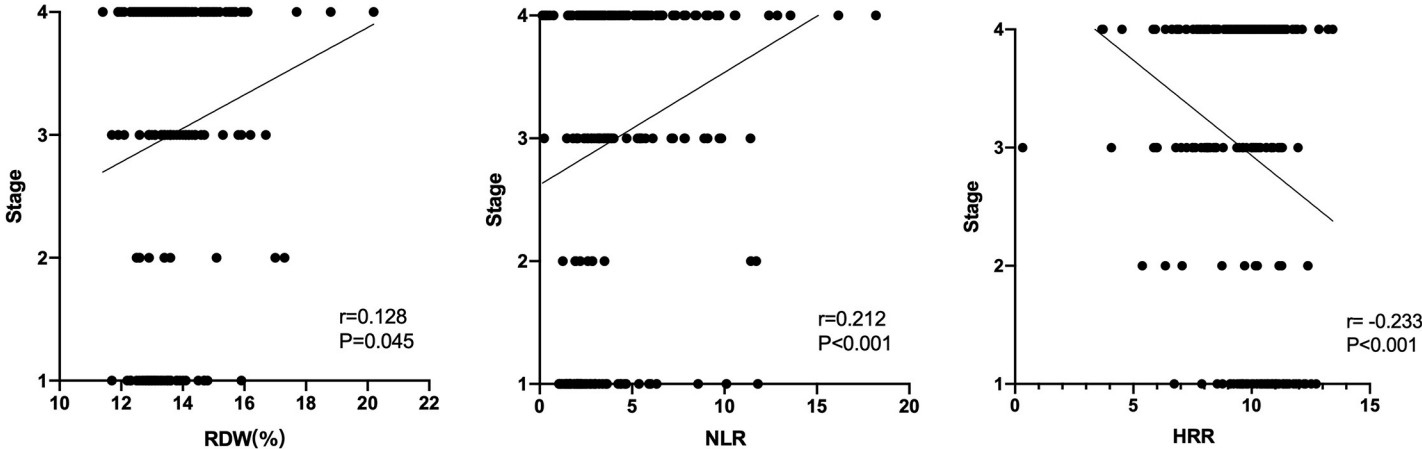

**Fig 3. A.** Correlation between RDW and TNM stage of NSCLC. RDW red blood cell distribution width; NSCLC non-small cell lung cancer. **B.** Correlation between RDW and TNM stage of NSCLC. RDW red blood cell distribution width; NSCLC non-small cell lung cancer. **C.** Correlation between HRR and TNM stage of NSCLC. HRR hemoglobin-to-red blood cell distribution width ratio; NSCLC non-small cell lung cancer.

higher tumor stage (p < 0.001), and poorer tumor differentiation (p < 0.001). In addition, apart from smoking history, HRR and age (p = 0.016), gender (p = 0.009), histology (p = 0.001), tumor size (p <0.001), lymph node metastasis (p <0.001), distant metastasis (p = 0.036), TNM stage (p <0.001), and differentiation degree (p <0.001) were significantly correlated.

### 4. Diagnostic value of RDW, NLR, HRR, and CEA in NSCLC

We used ROC curves to analyze and evaluate the diagnostic value (equal to the area under the curve, AUC) of individual and combined biomarkers (Table 4). Compared with NLR, HRR, and CEA, the diagnostic value of RDW was lower, with the AUC of 0.629 (95% CI: 0.565–0.692), p < 0.001, sensitivity and specificity of 63.83% and 57.55%, respectively. NLR and HRR were more sensitive to distinguish NSCLC patients from healthy volunteers (82.98% and 91.49%, respectively), while NLR and CEA were more specific (77.55% and 83.27%, respectively). When NLR and HRR were used in combination, the specificity of diagnosis increased significantly (93.60%). In addition, the combination of RDW with NLR, HRR, and CEA could show significantly higher diagnostic value compared with the use of any one marker alone, then the AUC was 0.925 (95% CI: 0.897–0.954), sensitivity and specificity were 79.60% and 93.60%, respectively.

### 5. Correlation analysis among the levels of RDW, NLR, and HRR

Correlation analysis showed that there was a very weak correlation between RDW and NLR (r = 0.160, p = 0.012; Fig 5A). In addition, we found a strong negative correlation between RDW and HRR (r = -0.667, p < 0.001; Fig 5B). Similarly, a negative correlation was found between HRR and NLR (r = -0.239, p < 0.001; Fig 5C).

## Discussion

The analysis of global cancer epidemiology has revealed that lung cancer is the main cause of cancer-related deaths worldwide [1]. Therefore, actively searching for relatively simple and rapid biomarkers related to early lung cancer diagnosis is of great importance. In this study, we investigated the relationship of RDW, NLR, and HRR with the clinical and biological

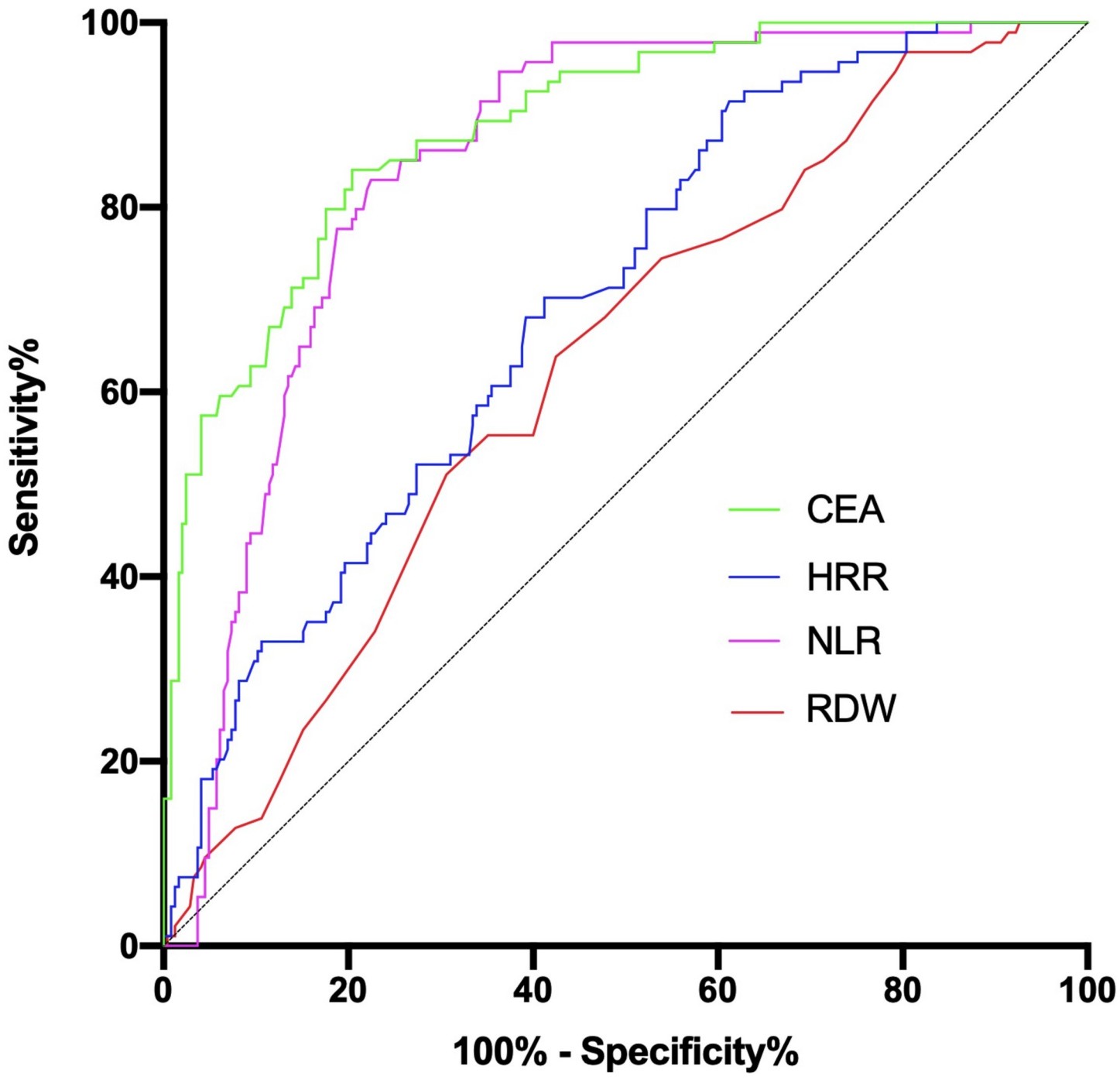

**Fig 4. The ROC curves for RDW, NLR, HRR and CEA.** Area under ROC curves were 0.629, 0.846, 0.696 and 0.887, respectively. P values were all <0.001. RDW red blood cell distribution width; NLR neutrophil-to-lymphocyte ratio; HRR hemoglobin-to-red blood cell distribution width ratio; CEA carcinoembryonic antigen; ROC receiver operating characteristic.

characteristics of NSCLC, and found that RDW, NLR, and HRR can be used as effective bio-markers for the diagnosis of NSCLC.

**Table 3. Correlation between clinicopathological features and RDW, NLR, and HRR in 245 all non-small cell lung cancers.**

| | RDW | | P value | NLR | | P value | HRR | | P value |
|---|---|---|---|---|---|---|---|---|---|
| | <13.25 | ≥13.25 | | <2.142 | ≥2.142 | | <9.481 | ≥9.481 | |
| Number | 104 | 141 | | 55 | 190 | | 95 | 150 | |
| **Age (years)** | | | | | | | | | |
| <64 | 53 | 43 | 0.002 | 30 | 66 | 0.012 | 28 | 68 | 0.016 |
| ≥64 | 51 | 98 | | 25 | 124 | | 67 | 82 | |
| **Gender** | | | | | | | | | |
| Male | 56 | 76 | >0.999 | 27 | 105 | 0.445 | 41 | 91 | 0.009 |
| Female | 48 | 65 | | 28 | 85 | | 54 | 59 | |
| **Smoking history** | | | | | | | | | |
| Yes | 52 | 79 | 0.367 | 25 | 106 | 0.219 | 45 | 86 | 0.149 |
| No | 52 | 62 | | 30 | 84 | | 50 | 64 | |
| **Histology** | | | | | | | | | |
| Squamous cell carcinoma | 37 | 70 | 0.037 | 20 | 87 | 0.222 | 55 | 52 | 0.001 |
| Adenocarcinoma | 67 | 71 | | 35 | 103 | | 40 | 98 | |
| **Tumor size** | | | | | | | | | |
| T1 | 31 | 30 | 0.470 | 24 | 37 | 0.002 | 8 | 53 | <0.001 |
| T2 | 39 | 63 | | 20 | 82 | | 45 | 57 | |
| T3 | 13 | 18 | | 5 | 26 | | 15 | 16 | |
| T4 | 21 | 30 | | 6 | 45 | | 27 | 24 | |
| **Lymph node metastasis** | | | | | | | | | |
| N0 | 45 | 45 | 0.169 | 32 | 58 | 0.002 | 18 | 72 | <0.001 |
| N1 | 8 | 7 | | 4 | 11 | | 4 | 11 | |
| N2 | 19 | 36 | | 8 | 47 | | 23 | 32 | |
| N3 | 32 | 53 | | 11 | 74 | | 50 | 35 | |
| **Distant metastasis** | | | | | | | | | |
| M0 | 53 | 64 | 0.438 | 38 | 79 | <0.001 | 37 | 80 | 0.036 |
| M1 | 51 | 77 | | 17 | 111 | | 58 | 70 | |
| **TNM stage** | | | | | | | | | |
| I-II | 38 | 32 | 0.022 | 31 | 39 | <0.001 | 12 | 58 | <0.001 |
| III-IV | 66 | 109 | | 24 | 151 | | 83 | 92 | |
| **Degree of differentiation** | | | | | | | | | |
| Well/moderated | 44 | 34 | 0.004 | 33 | 45 | <0.001 | 13 | 65 | <0.001 |
| Poorly | 60 | 107 | | 22 | 145 | | 82 | 85 | |

Data were expressed as number.

Binary logistic regression analysis with adjustment age and gender was used to control confounding factors.

*RDW* red blood cell distribution width; *NLR* neutrophil-to-lymphocyte ratio; *HRR* hemoglobin-to-red blood cell distribution width ratio.

P values were calculated by χ2 tests.

P<0.05 was considered significant.

In this retrospective study, we applied logistic regression to adjust the gender and age in the comparison of the medians of groups because we considered that the age and gender of each group as possible confounding factors. Our study found that RDW, NLR, and HRR can be used to distinguish patients with NSCLC from healthy volunteers. In addition, only the RDW in the NSCLC group with III-IV stage was significantly different from that in the benign pulmonary nodules group, while NLR and HRR could significantly distinguish patients with NSCLC and benign pulmonary nodules. We also found that NSCLC patients with III-IV stage

**Table 4. Diagnostic value of RDW, NLR, HRR and CEA in non-small cell lung cancers.**

|  | Cut-off | Sensitivity,% | Specificity,% | AUC | 95%CI |
|---|---|---|---|---|---|
| RDW | 13.25 | 63.83 | 57.55 | 0.629 | 0.565–0.692 |
| NLR | 2.14 | 82.98 | 77.55 | 0.846 | 0.804–0.888 |
| HRR | 9.48 | 91.49 | 38.78 | 0.698 | 0.637–0.755 |
| CEA | 2.01 | 76.60 | 83.27 | 0.887 | 0.849–0.924 |
| RDW+NLR | - | 74.70 | 88.30 | 0.849 | 0.808–0.891 |
| RDW+HRR | - | 38.40 | 91.50 | 0.695 | 0.636–0.754 |
| NLR+HRR | - | 70.20 | 93.60 | 0.852 | 0..811–0.893 |
| RDW+NLR+HRR | - | 72.70 | 88.30 | 0.851 | 0.810–0.892 |
| RDW+NLR+HRR+CEA | - | 79.60 | 93.60 | 0.925 | 0.897–0.954 |

*RDW* red blood cell distribution width; *NLR* neutrophil-to-lymphocyte ratio; *HRR* hemoglobin-to-red blood cell distribution width ratio; *CEA* carcinoembryonic antigen; *AUC* area under curve; *CI* confidence interval.

had significantly higher RDW and NLR and significantly lower HRR than patients with I-II stage. In addition, we divided patients with NSCLC into two groups on the basis of the optimal cut-off values of RDW, NLR, and HRR, and compared their relationship with the clinico-pathological characteristics of NSCLC. Our results showed that RDW was significantly correlated with age, histology, TNM stage, and differentiation degree. NLR was significantly correlated with age, tumor size, lymph node metastasis, distant metastasis, TNM stage, and differentiation degree. In addition to smoking history, HRR was significantly correlated with age, gender, histology, tumor size, lymph node metastasis, distant metastasis, TNM stage, and differentiation degree.

CEA, a classic tumor marker, can not only indicate the occurrence and development of lung cancer, but also be directly related to tumor infiltration and metastasis. CEA can well reflect the biological activities of tumor cells, such as proliferation, infiltration, invasion, migration and other capabilities [18–20]. However, we found that CEA had a high significance in the diagnosis of patients with benign pulmonary nodules, which has a certain interference effect in the diagnosis of lung cancer. Therefore, we considered combining multiple biomarkers to improve the effective diagnosis rate of NSCLC. Through the analysis of ROC curves, we

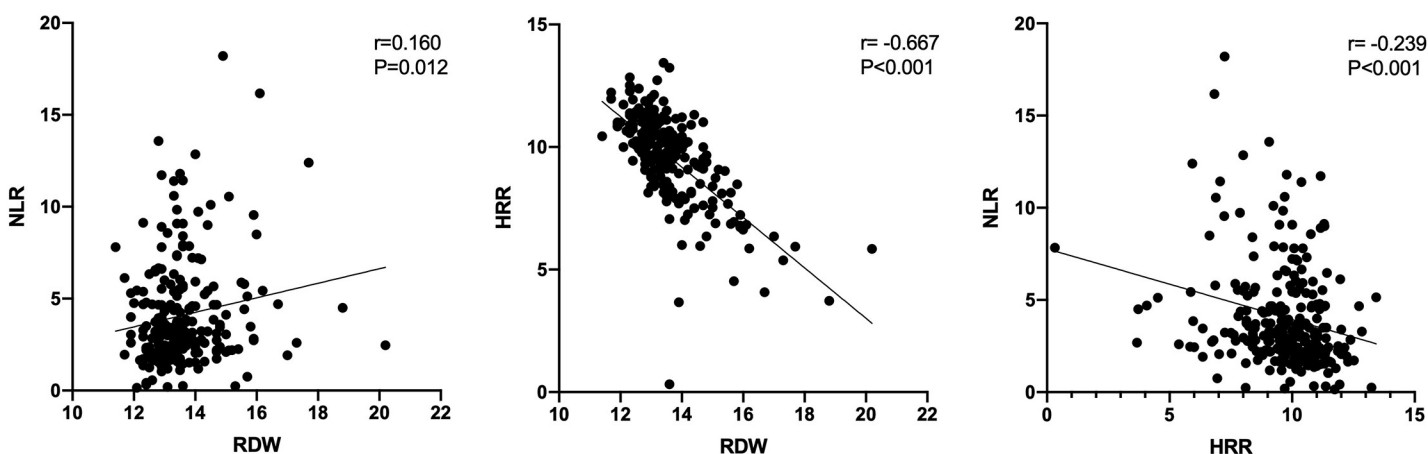

**Fig 5. A.** Correlation analysis of RDW with NLR. RDW red blood cell distribution width; NLR neutrophil-to-lymphocyte ratio. **B.** Correlation analysis of RDW with HRR. RDW red blood cell distribution width; HRR hemoglobin-to-red blood cell distribution width ratio. **C.** Correlation analysis of HRR with NLR. HRR hemoglobin-to-red blood cell distribution width ratio; NLR neutrophil-to-lymphocyte ratio.

found that the combination of RDW with NLR, HRR, and CEA could show significantly higher diagnostic value, larger AUC area, and higher sensitivity and specificity than any one marker alone. In addition, correlation analysis showed that there was a certain correlation among RDW, NLR, and HRR. Therefore, RDW, NLR and HRR can be used to diagnose NSCLC and assess progression well.

The "tumor microenvironment" concept can be traced back to the "seed and soil" hypothesis proposed by Stephen Paget in 1889 [21]. This hypothesis states that tumor cells can remodel the tumor microenvironment, and in turn tumor microenvironment remodeling can further affect the biological behavior of tumor cells. A growing body of evidence shows that the host's systemic response to tumor cells can cause inflammation, and systemic inflammation plays an indispensable role in the occurrence and development of various tumors [22]. On the one hand, systemic inflammatory response stimulates the immune process of the body, thereby inhibiting tumor development. On the other hand, it may promote the growth, malignancy and metastasis of tumor cells by disrupting the balance of the internal environment, suppressing immune response, inhibiting apoptosis and activating angiogenesis [23]. Our research on RDW, NLR, and HRR confirms the concept of "tumor microenvironment".

RDW is an evaluation index that reflects the changes in the sizes of the red blood cells involved in the circulatory response in peripheral blood. Similar to hemoglobin, RDW is often used to diagnose and differentiate types of anemia. Many recent studies have shown that RDW is closely related to the occurrence or prognosis of various cancers [12–14]. However, the exact mechanism of this connection has not been fully elucidated. Most studies have found that RDW is closely related to a variety of inflammatory markers, such as C-reactive protein, interleukin-6, and tumor necrosis factor. This relationship indicates that the cancer-mediated inflammatory microenvironment can lead to increased RDW expression in the body [24]. The inflammatory microenvironment may cause damage to iron metabolism in the body and inhibit erythropoietin production. These effects result in the entry of a large number of immature red blood cells from the bone marrow into the peripheral blood circulation. This phenomenon will eventually lead to variations in the sizes of the red blood cells involved in the circulatory response in the peripheral blood, as manifested by increased RDW levels [25]. Moreover, the increase in RDW may be related to oxidative stress response in the body [26]. Red blood cells, the most important oxygen-carrying medium in the body, will induce hypoxia or a hypoxic microenvironment when they differ in size or their functions are impaired. This microenvironment may accelerate the formation of tumor neovascularization by stimulating the production of vascular endothelial growth factor (VEGF) in the body [27]. Hypoxia or a hypoxic microenvironment can also induce the expression of the hypoxia-inducible factor (HIF) family, which may lead to tumor progression, cancer aggravation or poor prognosis [28]. VEGF and HIFs are widely involved in the tumor-mediated inflammatory microenvironment [29]. In addition, RDW has been found to be associated with malnutrition. The occurrence and development of cancer will cause the body to enter a cachexia state, which is characterized by the deficiency of iron, folic acid, and vitamin B12, and then affect the level of RDW in the body [30]. Therefore, RDW can reflect the systemic inflammatory response of cancer and the nutritional status of the body.

HRR, as a new type of biomarker, was first proposed to have predictive value for cancer by Sun et al. [15]. Sun et al. found that there was no significant difference in the overall survival period of patients with esophageal squamous cell carcinoma when they used hemoglobin and RDW as prognostic factors. However, they found that there was a significant association between HRR and the survival outcomes of patients with esophageal squamous cell carcinoma. Thus, HRR can be used as an important prognostic factor that is unaffected by other risk factors. Our study found that RDW had significant difference only between the NSCLC group

with III-IV stage and the benign pulmonary nodule group, whereas HRR could be used to significantly distinguish patients with NSCLC and patients with benign pulmonary nodules. Moreover, HRR had higher diagnostic value and higher sensitivity than RDW. Therefore, we believe that HRR has great importance in the diagnosis of NSCLC. In addition, both hemoglobin and RDW may be affected by various diseases other than cancer [15–17], while HRR can minimize the potential effects, and theoretically reflect the systemic inflammatory response and nutritional status of the body. Therefore, HRR can be used as a relatively more reliable biomarker in the assessment of cancer progression.

NLR reflects the quantitative relationship between neutrophils and lymphocytes in the body. Neutrophils are derived from bone marrow stem cells, and account for about 50% - 70% of the total number of leukocytes in the peripheral blood. They are considered as the main immune cells that protect the body from microbial infection and eliminate pathogens [31]. A large number of studies have shown that under the interaction of tumor cells and the tumor microenvironment, neutrophils can reshape their own phenotype and function, and then participate in the occurrence and development of tumor through a variety of mechanisms, including promoting the proliferation, migration and invasion capabilities of tumor cells, and the formation of new blood vessels in tumor cells [8,32]. In contrast to neutrophils, lymphocytes play a key role in the body's protective immunity by inhibiting the proliferation and migration of tumor cells. Lymphocytes can induce the death of cytotoxic cells, and produce cytokines, such as interleukin-4 and interleukin-12, that inhibit the proliferation and metastasis of tumor cells [33,34]. Therefore, it is believed that the reduction in the number of peripheral blood lymphocytes may lead to the attenuation or disappearance of the immune response to tumor, and thus promote the occurrence and development of tumor. All in all, NLR can be used as a reliable indicator to reflect the systemic inflammatory response in tumor progression in vivo.

This study has several limitations. First, it involved retrospective analysis, which included the manual extraction and input of clinical data. Although there was no missing or erroneous data for laboratory values or basic patient information in our analysis, patient selection bias is a possible risk. Second, this study did not evaluate various potential confounding factors, such as severe infections and local ischemia, that may affect peripheral blood cell counts. Thirdly, due to some constraints, the effect of age and smoking on the indicators was not fully evaluated in this study. Therefore, a large-scale prospective study is warranted to confirm the results of the current study.

## Conclusions

In conclusion, RDW, NLR, and HRR can be used as simple, cheap, and effective biomarkers for diagnosing and evaluating the progress of patients with NSCLC in clinical practice, which will be helpful for clinicians to select the appropriate clinical treatment options and predict disease prognosis.

## Supporting information

**S1 Data.**
(XLSX)

## Author Contributions

**Conceptualization:** Jin-liang Chen, Jin-nan Wu.

**Data curation:** Jin-nan Wu.

**Formal analysis:** Jin-nan Wu.

**Investigation:** Jin-liang Chen, Jin-nan Wu, Xue-dong Lv.

**Methodology:** Jin-liang Chen.

**Project administration:** Jian-rong Chen.

**Software:** Jin-liang Chen, Jin-nan Wu, Dong-mei Zhang.

**Supervision:** Qi-chang Yang.

**Validation:** Dong-mei Zhang.

**Visualization:** Xue-dong Lv.

**Writing – original draft:** Jin-nan Wu.

**Writing – review & editing:** Jian-rong Chen.

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
