## [Decision Letter · Decision Letter 0]

13 Jul 2020

PONE-D-20-13045

The value of red blood cell distribution width, neutrophil-to-lymphocyte ratio, and hemoglobin-to-red blood cell distribution width ratio in the progression of non-small cell lung cancer

PLOS ONE

Dear Dr. Wu,

Thank you for submitting your manuscript to PLOS ONE. After careful consideration, we feel that it has merit but does not fully meet PLOS ONE’s publication criteria as it currently stands. Therefore, we invite you to submit a revised version of the manuscript that addresses the points raised during the review process.

Normally I require at least two reviewers; however, given the desire for timely turn-around, I decided based on one review.  Please address all of the issues raised.

We look forward to receiving your revised manuscript.

Kind regards,

Jeffrey Chalmers, Ph.D.

Academic Editor

PLOS ONE

Journal Requirements:

2. In the ethics statement in the manuscript and in the online submission form, please provide additional information about the patient records used in your retrospective study, including: a) whether all data were fully anonymized before you accessed them; b) the date range (month and year) during which patients' medical records were accessed. If patients provided informed written consent to have data from their medical records used in research, please include this information.

4. Please upload a copies of Figures 1 to 5, to which you refer in your text. If the figures are no longer to be included as part of the submission please remove all reference to them within the text.

Reviewers' comments:

Reviewer's Responses to Questions

**Comments to the Author**

1. Is the manuscript technically sound, and do the data support the conclusions?

Reviewer #1: Partly

2. Has the statistical analysis been performed appropriately and rigorously? 

Reviewer #1: Yes

3. Have the authors made all data underlying the findings in their manuscript fully available?

Reviewer #1: Yes

4. Is the manuscript presented in an intelligible fashion and written in standard English?

Reviewer #1: Yes

5. Review Comments to the Author

Reviewer #1: This manuscript reports the value of several hematologic parameters in the progression of non-small cell lung cancer (NSCLC). Red blood cell distribution width (RDW), neutrophil-to-lymphocyte ratio (NLR), and hemoglobin-to-red blood cell distribution (HRR) values were evaluated in 245 patients with NSCLC, and compared to 97 patients with benign nodules and to 94 healthy volunteers. Results showed that RDW, NLR, and HRR could be used to distinguish patients with NSCLC from healthy controls. Furthermore, NLR and HHR could be used to distinguish between patients with NSCLC and benign nodules, while RDW can be used for distinguishing the NSCLC group at III-IV stage. As these 3 parameters are available in any hematology analyzer, and are fast and cheap to obtain, the information reported in this study can be very useful for those studying new diagnostic tools to predict cancer progression and prognosis. Therefore, I think this manuscript merits publication in Plos One. In the following, authors can find minor comments that can help to improve the quality of the manuscript:

- Page 10: “The healthy volunteers had no acute and chronic infectious diseases; no vital organ diseases. such as heart, liver, kidney and digestive system diseases”. Please, remove point before “such”.

- Tables 1, 2 and 3. Please correct the typographical error in “smoking history”.

- Page 12: “Leukocyte count, neutrophil count, lymphocyte count, hemoglobin, and CEA in patients with NSCLC were significantly higher than those in patients with benign lung nodules and healthy controls”. Please, correct the sentence. Lymphocyte count and hemoglobin are lower among patients with NSCLC, which is the primary reason why NLR is high and HRR is low for these patients.

- I recommend the authors to include either the percentage of patients with abnormal hematological parameters in Table 2, or to include the normal values of these parameters in the table (as another column titled normal range).

- Hemoglobin normal range varies with gender and smoking status. Proof of this is that the only parameter correlated to smoking history and gender is HRR (none of the other 2 parameters are correlated to these clinical characteristics). I wonder if the authors can safely apply this parameter as a diagnostic tool independently of gender and smoking history. Please, discuss this issue.

6. PLOS authors have the option to publish the peer review history of their article (what does this mean?). If published, this will include your full peer review and any attached files.

Reviewer #1: No

---

## [Author Response · Author response to Decision Letter 0]

3 Aug 2020

Dear editors:

Thank you very much for your letter and the comments from the referees about our paper submitted to PLOS ONE (PLOS ONE manuscript number: PONE-D-20-13045).We have checked the manuscript and revised it according to the comments. We submit here the revised manuscript as well as a list of changes.

Part I Responses to Editor

1.Please ensure that your manuscript meets PLOS ONE’s style requirements, including those for file naming. 

Reply: We have made modifications in strict accordance with the PLOS ONE’s style requirements. If there are any errors, please don’t hesitate to let us know.

2.In the ethics statement in the manuscript and in the online submission form, please provide additional information about the patient records used in your retrospective study, including: a) whether all data were fully anonymized before you accessed them; b) the date range (month and year) during which patients' medical records were accessed. If patients provided informed written consent to have data from their medical records used in research, please include this information.

Reply: We can provide additional information about the patient records used in our retrospective study, including: a) all data were fully anonymized before we accessed them; b) the date range (month and year) during which patients' medical records were accessed was June 2016 to June 2019. In addition, we can provide the patients’ informed written consent to have data from their medical records used in research. We have uploaded the informed written consent and related materials of patients in the form of attachment.

3.PLOS only allows data to be available upon request if there are legal or ethical restrictions on sharing data publicly. 

Reply: All relevant data are within the manuscript and its Supporting Information files.

4.Please upload a copies of Figures 1 to 5, to which you refer in your text. 

Reply: We have uploaded a copies of Figures 1 to 5.

Part II Responses to Referee 1

1.Page 10: “The healthy volunteers had no acute and chronic infectious diseases; no vital organ diseases. such as heart, liver, kidney and digestive system diseases”. Please, remove point before “such”.

Reply: We have revised the description of this sentence. Now the sentence is “The healthy volunteers had no acute and chronic infectious diseases; no vital organ diseases; and no genetic family tumor history”.

2.Tables 1, 2 and 3. Please correct the typographical error in “smoking history”.

Reply: We have corrected the typographical error in “smoking history”.

3.Page 12: “Leukocyte count, neutrophil count, lymphocyte count, hemoglobin, and CEA in patients with NSCLC were significantly higher than those in patients with benign lung nodules and healthy controls”. Please, correct the sentence. Lymphocyte count and hemoglobin are lower among patients with NSCLC, which is the primary reason why NLR is high and HRR is low for these patients.

Reply: We have revised the description of this sentence. Now the sentence is “Leukocyte count, neutrophil count, and CEA in patients with NSCLC were significantly higher than those in patients with benign lung nodules and healthy controls”.

4.Recommend the authors to include either the percentage of patients with abnormal hematological parameters in Table 2, or to include the normal values of these parameters in the table (as another column titled normal range).

Reply: We have include the normal values of these parameters in the table (as another column titled normal range).

5.Hemoglobin normal range varies with gender and smoking status. Proof of this is that the only parameter correlated to smoking history and gender is HRR (none of the other 2 parameters are correlated to these clinical characteristics). I wonder if the authors can safely apply this parameter as a diagnostic tool independently of gender and smoking history. Please, discuss this issue.

Reply: In this article, we divided the patients with NSCLC into two groups in accordance with the optimal cut-off value of HRR, and then analyzed the relationship of HRR with the clinicopathological characteristics of the patients. Therefore, apart from smoking history, HRR and age (p = 0.016), gender (p = 0.009), histology (p = 0.001), tumor size (p <0.001), lymph node metastasis (p <0.001), distant metastasis (p = 0.036), TNM stage (p <0.001), and differentiation degree (p <0.001) were significantly correlated.

Then we considered that the age and gender of each group as possible confounding factors, so we applied logistic regression to adjust the gender and age in the comparison of the medians of groups. And then We divided 245 NSCLC patients into smoking group and non-smoking group according to smoking history, and compared the difference of HRR between the two groups: 9.921(8.593-10.92) vs 9.816(8.694-10.70), p = 0.4462, indicated no significant difference. Then we divided 245 NSCLC patients into male group and female group according to gender, and compared the difference of HRR between the two groups: p > 0.05, indicated no significant difference. We have had to admit that due to some constraints, the effect of age, gender, and smoking on the indicators was not fully evaluated in this study. Therefore, a large-scale prospective study is warranted to confirm the results of the current study.

All the lines and pages indicated above are in the revised manuscript.

Thank you and all the reviewers for the kind advice. If you have any question about this paper, please don’t hesitate to let me know.

Thank you and best regards.

Yours sincerely,

Jinnan Wu.

---

## [Editor Report · Decision Letter 1]

6 Aug 2020

The value of red blood cell distribution width, neutrophil-to-lymphocyte ratio, and hemoglobin-to-red blood cell distribution width ratio in the progression of non-small cell lung cancer

PONE-D-20-13045R1

Dear Dr. Wu,

We’re pleased to inform you that your manuscript has been judged scientifically suitable for publication and will be formally accepted for publication once it meets all outstanding technical requirements.

Kind regards,

Jeffrey Chalmers, Ph.D.

Academic Editor

PLOS ONE
---

## [Editor Report · Acceptance letter]

12 Aug 2020

PONE-D-20-13045R1 

The value of red blood cell distribution width, neutrophil-to-lymphocyte ratio, and hemoglobin-to-red blood cell distribution width ratio in the progression of non-small cell lung cancer 

Dear Dr. Wu:

I'm pleased to inform you that your manuscript has been deemed suitable for publication in PLOS ONE. Congratulations! Your manuscript is now with our production department. 

Kind regards, 

on behalf of

Dr. Jeffrey Chalmers 

Academic Editor

PLOS ONE